# LLMAAA: Making Large Language Models as Active Annotators

**Ruoyu Zhang[1,*], Yanzeng Li[1], Yongliang Ma[2], Ming Zhou[2], Lei Zou[1]**

[1]Wangxuan Institute of Computer Technology, Peking University, Beijing, China

[2]Langboat Technology, Beijing, China

{ry_zhang, zoulei}@pku.edu.cn, liyanzeng@stu.pku.edu.cn,

{mayongliang, zhouming}@langboat.com

## Abstract

Prevalent supervised learning methods in natural language processing (NLP) are notoriously data-hungry, which demand large amounts of high-quality annotated data. In practice, acquiring such data is a costly endeavor. Recently, the superior performance of large language models (LLMs) has propelled the development of dataset generation, where the training data are solely synthesized from LLMs. However, such an approach usually suffers from low-quality issues and requires orders of magnitude more labeled data to achieve satisfactory performance. To fully exploit the potential of LLMs and make use of massive unlabeled data, we propose LLMAAA, which takes LLMs as annotators and puts them into an active learning loop to determine what to annotate efficiently. To learn robustly with pseudo labels, we optimize both the annotation and training processes: (1) we draw $k$-NN samples from a small demonstration pool as in-context examples, and (2) we adopt the automatic reweighting technique to assign training samples with learnable weights. Compared with previous approaches, LLMAAA features both **efficiency** and **reliability**. We conduct experiments and analysis on two classic NLP tasks, named entity recognition and relation extraction. With LLMAAA, task-specific models trained from LLM-generated labels can outperform their teacher LLMs within only hundreds of annotated examples, which is much more cost-effective than other baselines[1].

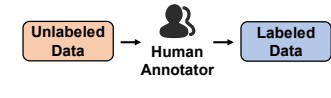

(a) Human annotation as supervision.

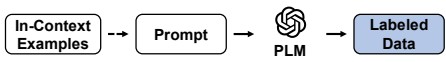

(b) Text generation as supervision.

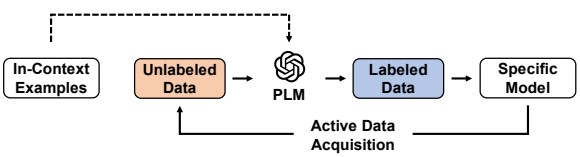

(c) LLMAAA: Active LLM annotation as supervision.

Figure 1: Comparing LLMAAA with other frameworks. We actively acquire annotations from LLM for efficiency, requiring little human effort.

## 1 Introduction

Large language models (LLMs) have exhibited remarkable few-shot performance in a wide range of tasks, with only a few demonstrations and well-designed prompts (Brown et al., 2020; Ding et al., 2022; Liu et al., 2023). However, with rapid advancements comes vast potential risks in adopting LLMs for widespread downstream production applications. One of the main concerns is about data privacy and security. Under the prevalent "Language-Model-as-a-Service" (LMaaS, Sun et al., 2022) setting, users are required to feed their own data, potentially including sensitive or private information, to third-party LLM vendors to access the service, which increases the risk of data leakage (Lyu et al., 2020; Yu et al., 2022; Li et al., 2023). Besides, LLMs usually consume abundant tokens by continuous requests to APIs, where the marginal cost and latency become substantial in large-scale or real-time applications, hindering LLMs' practical deployment in cost-sensitive scenarios (Goyal et al., 2020; Cao et al., 2023).

On the other hand, training task-specific models (TAMs) for NLP tasks necessitates extensive amounts of labeled data. Due to the superior generative capacity of LLMs, some researchers attempt to synthesize training data with text generation (Meng et al., 2022; Ye et al., 2022), as depicted

---

*This work was done during an internship at Langboat Technology.

[1]Our code and data are available at https://github.com/ridiculouz/LLMAAA.

in Figure 1. However, the generated text usually struggles with low-quality issues and may exhibit domain shifts with test data (Gao et al., 2023). To exploit the abundant unlabeled corpus, an alternative is to employ LLMs as annotators, which generate labels in a zero-shot or few-shot manner. While this approach seems promising, it is important to acknowledge that LLM-generated labels inevitably contain noise, especially when applied to challenging tasks and domain-specific data (Agrawal et al., 2022; Kazemi et al., 2023). Besides, larger models come with heavier expenses, and it is also crucial to reduce the annotation cost when the budget is restricted.

To enhance the **reliability** (i.e. accuracy) of TAMs' performance as well as to ensure the data **efficiency** in annotation cost, we propose LLMAAA, an innovative framework that integrates active learning into the LLM annotation process, i.e., making LLMs as Active Annotators. By exploring different active acquisition strategies, LLMAAA enables the LLM to annotate more informative instances that benefit model performance more. To train TAMs reliably, we optimize both the annotation and training processes within LLMAAA framework. Firstly, we employ prompt engineering techniques to enhance LLMs' performance by (1) selecting $k$-NN samples from a demonstration pool as in-context examples, and (2) building fine-level descriptions aligned with natural language for unnatural labels (e.g., category labels in the RE task). The valuable contextual information helps improve the quality of LLM annotations substantially. During training, we adopt the automatic reweighting technique (Ren et al., 2018) to assign learnable weights to the *silver*[2] training samples. This strategy allows the model to prioritize more informative and representative samples while simultaneously reducing the impact of noisy annotations.

We evaluate LLMAAA on two practical NLP tasks: named entity recognition (NER) and relation extraction (RE). Experiments show that: (1) with small-scale *gold* data (~100 examples) serving for demonstration and validation, the trained TAMs can outperform their teacher LLMs within hundreds of *silver* samples via LLMAAA; (2) our approach is significantly more data efficient compared to prevalent data generation methods, which

usually require large-scale synthetic training data (size varying from 10k to 200k, Ye et al., 2022; Gao et al., 2023). These results confirm the potential of LLMAAA as a practical and cost-efficient solution to make LLMs as *good* annotators. The TAMs created through our framework offer advantages in terms of task-specific performance, data privacy, and inference costs, which release the capacity of LLMs for real-world productivity.

We summarize our contributions as follows:

- We propose LLMAAA, a framework to employ LLMs as annotators, featuring both efficiency and reliability.

- LLMAAA is capable to train TAMs that outperform teacher LLMs within hundreds of annotated samples, on classic NLP tasks like NER and RE.

- LLMAAA sheds light on the practical substitution of LLMs, with a cost-effective, privacy-ensured, yet well-performing solution.

## 2 Related Work

**LLM and In-Context Learning** Large language models (LLMs), usually pretrained on large-scale corpus to capture rich linguistic patterns and generate coherent text (Brown et al., 2020; Raffel et al., 2020; Chowdhery et al., 2022; OpenAI, 2023; Touvron et al., 2023), have shown remarkable performance in a wide range of NLP tasks (Min et al., 2021; Zhao et al., 2023). With the proposal of in-context learning (Brown et al., 2020), prompt engineering has been extensively explored to steer LLMs' behavior for desired outcomes. These techniques design specific prompts or instructions to guide models' outputs (Ding et al., 2022; Liu et al., 2023), either in rule-based (Shin et al., 2020) or learning-based (Lester et al., 2021) manners. Recent trend focuses on the strong reasoning capabilities of LLMs and enhances LLMs' performance on complex task with chain-of-thought (CoT) prompting (Wei et al., 2023a). In general, prompt engineering improves the controllability and performance of LLMs in few-shot and zero-shot settings (Zhong et al., 2021), and enables LLMs to solve specific tasks, e.g. information extraction (Wei et al., 2023b; Wang et al., 2023).

**Dataset Synthesis** Supervised learning methods in NLP are often limited by high-quality annotated data. To address the bottleneck, researchers have explored techniques to synthesize training

---

[2]We refer *gold* data to ground-truth/human-labeled samples, and *silver* data to LLM-labeled samples.

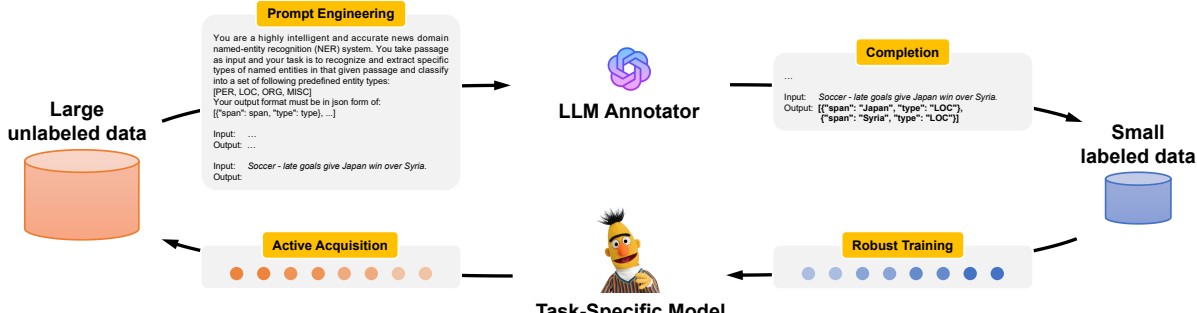

Figure 2: LLMAAA puts the LLM annotator in an active learning iteration, which mainly consists of three novel components: (1) an LLM annotator optimized with prompt engineering that generates pseudo labels, (2) an active acquisition mechanism for efficient data selection, and (3) an automatic reweighting technique to ensure robust learning with noisy labels. The annotation and training stages run iteratively and gradually produce labeled data for task-specific models.

data with LLMs, either by annotation or by generation. Following the first line of research, Feng et al. (2021); Chen et al. (2023) employ LLMs as unsupervised annotators to generate dialogue datasets. Recently, AnnoLLM (He et al., 2023) makes LLMs' performance on par with crowd-source annotators by chain-of-thought prompting and self-generated explanations. As a contemporary work, Bansal and Sharma (2023) use LLMs for annotation in the domain transfer setting. Under the formulation of active learning, they propose a new metric, conditional informativeness, that works well with noisy labels. Among generation-based methods, Wang et al. (2021) first use LLM with few-shot prompts to generate training data. Schick and Schütze (2021) attempt to generate labeled text counterparts and text pairs for semantic textual similarity tasks. ZEROGEN (Ye et al., 2022) and SUNGEN (Gao et al., 2023) further extend this practice to zero-shot learning by training small models with zero-shot LLM-generated datasets. However, these approaches still suffer the low-quality and domain-shift issues of the synthetic data, and *none* of them consider the cost efficiency of data generation via LLMs.

**Labor Efficiency and Active Learning** Active learning is a technique proposed to minimize the annotation cost during the labeling process (Settles, 2009; Ren et al., 2021). A popular setting for active learning is the pool-based paradigm, which aims to select the most beneficial samples from an unlabeled data pool based on criteria including uncertainty (Lewis and Gale, 1994; Houlsby et al., 2011; Gal et al., 2017), diversity (Huang et al., 2010; Sener and Savarese, 2018), and hybrid ob-

jectives (Du et al., 2017; Yang et al., 2017; Ash et al., 2020; Margatina et al., 2021). The selected samples are annotated by human annotators and then added into the labeled dataset iteratively.

## 3 LLM as Active Annotator

To exploit LLMs' superior few-shot performance and leverage abundant unlabeled data, we attempt to take LLM as annotator and train task-specific models for inference. An ideal process should be both **efficient** and **reliable**: we want to learn TAMs robustly with minimal LLM-generated labels.

Concretely, our solution is to make LLMs as Active Annotator. As shown in Figure 2, LL-MAAA comprises three key components: (1) an LLM annotator that generates pseudo labels of given data, (2) an active acquisition mechanism for efficient data selection, and (3) an automatic reweighting technique to ensure robust learning with noisy labels. LLMAAA iterates the three stages to gradually produce stronger TAMs.

### 3.1 Optimizing LLM as Better Annotator

In-context learning (i.e. PROMPTING) enables LLM to conduct few-shot inference without fine-tuning. Given a manually-designed prompt $T(\cdot, \cdot)$, a demonstration set $\mathcal{S} = \{\mathbf{x}^i, y^i\}_{i=1}^k$ and the query example $\mathbf{x}_q$, PROMPTING first builds a sentence $T(\mathcal{S}, \mathbf{x}_q)$, conditioned on which LLM then generates a text sequence

$$\mathbf{y}_q = \underset{\mathbf{y}}{\arg\max}\, P_{LM}(\mathbf{y}|T(\mathcal{S}, \mathbf{x}_q)).$$

Finally, $\mathbf{y}_q$ is mapped to the label space $\mathcal{Y}$.

Despite the decent abilities, previous studies show that the design of task-specific prompts has

a large impact on performance, varying between near state-of-the-art and random guess (Gao et al., 2021; Lu et al., 2022b). Finding the *best* prompts for given tasks and given data points is intractable. However, there are several principles turn out to be effective, compared with plain instruction.

$k$-**NN Example Retrieval** To select good in-context examples, Liu et al. (2022) propose a $k$-NN retrieval strategy, which first embeds the demonstration pool $\mathcal{D}_{\text{demo}}$ and query sample to vector representations, and then retrieves the nearest $k$ neighbors of the query to form its exemplars. The rationale behind this is that semantically similar examples may help LLM answer the query better. Following their practice, we use Sentence-BERT (Reimers and Gurevych, 2019, 2020) to build the representations.

**Label Verbalizer** In classification tasks, the surface forms of labels may induce difficulties and ambiguities. Taking relation classification for instance, the label "per:parents" can indicate either "subject is the parent of object" or "object is the parent of subject", depending on its definition. To address this problem, we utilize a label verbalizer to transform the surface forms to natural language descriptions with pre-defined templates (Sainz et al., 2021; Lu et al., 2022a), serving as fine-level guidance. The semantic templates we use are shown in Table 7.

## 3.2 Active Data Acquisition

Active learning (AL) seeks to reduce labeling efforts by strategically choosing which examples to annotate. We consider the standard *pool-based* setting, assuming that a large pool of unlabeled data $\mathcal{D}_{\text{pool}}$ is available. AL loop starts with a seed labeled set $\mathcal{D}_{\text{labeled}}$. At each iteration, we train a model $M$ on $\mathcal{D}_{\text{labeled}}$ and then use acquisition function $f(\cdot, M)$ to acquire a batch $\mathcal{B}$ consisting of $b$ examples from $\mathcal{D}_{\text{pool}}$. We then query the LLM annotator to label $\mathcal{B}$. The labeled batch is then removed from the pool $\mathcal{D}_{\text{pool}}$ and added to labeled set $\mathcal{D}_{\text{labeled}}$, and will serve as training data for the next iteration. The process is repeated for $t$ times.

Active acquisition strategies generally maximize either *uncertainty* or *diversity*. On one hand, uncertainty-based methods leverage model predictions to select *hard* examples. On the other hand, diversity-based methods exploit the heterogeneity of sampled data. We will cover some common

strategies for thorough comparisons, and illustrate with classification task for simplicity[3].

**Random** We consider random selection as baseline, which samples uniformly from $\mathcal{D}_{\text{pool}}$. Typically pool data and test data share the same distribution, thus the sampled batch is expected to be i.i.d. with test data.

**Maximum Entropy** Entropy is one of the most widely used estimations of uncertainty (Settles, 2009). Data for which the model $M$ has the highest entropy are sampled for annotation according to

$$\operatorname*{argmax}_{\mathbf{x} \in \mathcal{D}_{\text{pool}}} - \sum_{y \in \mathcal{Y}} P_M(y|\mathbf{x}) \log P_M(y|\mathbf{x}).$$

**Least Confidence** Culotta and McCallum (2005) propose to sort examples with the probability assigned by $M$ to predicted class $\hat{y}$, which samples

$$\operatorname*{argmax}_{\mathbf{x} \in \mathcal{D}_{\text{pool}}} \left(1 - P_M(\hat{y}|\mathbf{x})\right).$$

$K$-**Means** Diversity sampling intends to select batches of data that is heterogeneous in the feature space. Following Yuan et al. (2020), we apply $k$-means clustering to the $l_2$-normalized embeddings of $M$[4], and sample the nearest neighbors of the $k$ cluster centers.

## 3.3 Robust Learning with Noisy Labels

LLM annotators inevitably produce noisy labels, especially with harder tasks and domain-specific data. To stay robust against training label bias, we adopt the automatic reweighting technique (Ren et al., 2018) to assign different weights to training examples adaptively.

We assume that a small-scale validation set $\mathcal{D}_{\text{val}}$ with clean labels (e.g. human annotations) is available throughout learning, with $|\mathcal{D}_{\text{val}}| \ll |\mathcal{D}_{\text{pool}}|$. Concisely, automatic reweighting learns sample weights $\mathbf{w}$ by a meta-learning objective that minimizes validation loss w.r.t. $\mathbf{w}$, and uses online approximation to eliminate the nested loop of optimization. The training process of TAM is shown in Algorithm 1.

## 4 Tasks

We instantiate LLMAAA with two tasks: named entity recognition (NER) and relation extraction

---

[3]Adaptation to other settings (e.g. sequence tagging) will be introduced in § 4.

[4]We use BERT family as $M$'s encoder, and the embeddings refer to BERT output.

**Algorithm 1:** Automatic Reweighting

**Input:** Noisy data $\mathcal{D}_{\text{train}}$, clean data $\mathcal{D}_{\text{val}}$, batch size $n, m$, initial parameter $\theta_0$, step $S$

**Output:** Trained parameter $\theta_S$

**for** $s = 0, \ldots, S - 1$ **do**
$\quad \mathcal{B}_{\text{train}} \leftarrow \text{SampleBatch}(\mathcal{D}_{\text{train}}, n)$
$\quad \mathcal{B}_{\text{val}} \leftarrow \text{SampleBatch}(\mathcal{D}_{\text{val}}, m)$
$\quad \{\hat{y}^i_{\text{train}}\}^n_{i=1} \leftarrow \text{Forward}(\mathcal{B}_{\text{train}}, \theta_s)$
$\quad$ // build computation graph with automatic diffErtiation
$\quad \epsilon \leftarrow 0; l_{\text{train}} \leftarrow \sum^n_{i=1} \epsilon_i \mathcal{L}(y^i_{\text{train}}, \hat{y}^i_{\text{train}})$
$\quad \nabla\theta_s \leftarrow \text{Backward}(l_{\text{train}}, \theta_s)$
$\quad \hat{\theta}_s \leftarrow \theta_s - \alpha\nabla\theta_s$
$\quad \{\hat{y}^i_{\text{val}}\}^m_{i=1} \leftarrow \text{Forward}(\mathcal{B}_{\text{val}}, \hat{\theta}_s)$
$\quad l_{\text{val}} \leftarrow \frac{1}{m} \sum^m_{i=1} \mathcal{L}(y^i_{\text{val}}, \hat{y}^i_{\text{val}})$
$\quad \nabla\epsilon \leftarrow \text{Backward}(l_{\text{val}}, \epsilon)$
$\quad$ // truncate weights to zero, and normalize to one
$\quad \tilde{w} \leftarrow \max(-\nabla\epsilon, 0);$
$\quad w \leftarrow \frac{\tilde{w}}{\sum_j \tilde{w} + \delta(\sum_j \tilde{w})}$
$\quad \hat{l}_{\text{train}} \leftarrow \sum^n_{i=1} w_i \mathcal{L}(y^i_{\text{train}}, \hat{y}^i_{\text{train}})$
$\quad \nabla\theta_s \leftarrow \text{Backward}(\hat{l}_{\text{train}}, \theta_s)$
$\quad \theta_{s+1} \leftarrow \text{OptimizerStep}(\theta_s, \nabla\theta_s)$

(RE). We opt for two simple yet effective models as TAMs, and leave other choices for future study.

### 4.1 Named Entity Recognition

**Formulation** NER aims to extract entities $\{e_i\}$ from text $\mathbf{x}$, where $e_i$ can be expressed as a continuous span of sequences with predefined type. We consider the flat scenario (i.e. no overlapping entities), in which NER can be reformulated as a sequence tagging problem with `BIO` label.

To smoothly adapt uncertainty-based active functions from classification task to sequence tagging, we provide three pooling options: average, sum, and max. In practice, we adopt average and sum operations for better empirical performance.

**Model** Following Devlin et al. (2019), we leverage BERT to convert tokens into vectorized features, and use a linear classifier with activation to predict the `{class}-BIO` label for each token.

### 4.2 Relation Extraction

**Formulation** Given subject entity $e_{\text{subj}}$ and object entity $e_{\text{obj}}$ in a sentence, RE classifies their relation into a predefined set $\mathcal{R} \cup \{\text{NA}\}$.

**Model** We use the same model architecture as Baldini Soares et al. (2019), which first encloses entity spans with special tokens `[E]` and `[\E]`, then encodes the sentence with BERT. The concatenated embedding of subject and object is fed into a linear classifier with activation for final prediction.

## 5 Experiments and Analysis

### 5.1 Setup

**Dataset** We experiment with three different NLP datasets: Chinese OntoNotes 4.0 (Weischedel et al., 2011) and English CoNLL03 (Tjong Kim Sang and De Meulder, 2003) for NER, and Re-TACRED (Stoica et al., 2021) for RE. For Re-TACRED, we select a subset describing personal relationships and balance the NA relation instances to the original portion. Details of dataset statistics are described in Appendix A. We report the precision, recall, and micro F1 for both tasks.

**Baselines** We compare LLMAAA with the following baselines: (1) PROMPTING. The prompt-based direct inference on test data, using the same engineering techniques as LLMAAA's teacher LLMs. (2) SUPERVISED. The TAMs are trained on clean-labeled data $\mathcal{D}_{\text{val}}$ used in LLMAAA's demonstration/validation. (3) ZEROGEN (Ye et al., 2022). Zero-shot data synthesis method via text generation. (4) FEWGEN. A data synthesis method that enhances ZEROGEN with in-context examples uniformly sampled from the demonstration pool.

**Implementation** We use ChatGPT[5] as LLM annotator for main experiments, and adopt BERT (Devlin et al., 2019; Cui et al., 2021) as TAM's encoder. We also explore with other LLM annotators, GPT-3 (Brown et al., 2020) and GPT-4 (OpenAI, 2023), in § 6. We randomly sample 100 examples from the original validation sets as *gold* data, reusing the same set for demonstration $\mathcal{D}_{\text{demo}}$ and validation $\mathcal{D}_{\text{val}}$. We use the original training sets as $\mathcal{D}_{\text{pool}}$ and randomly initialize seed labeled set $\mathcal{D}_{\text{labeled}}$ with a size of 50 and acquire 50 samples per batch for 9 iterations, which generates 500 *silver* annotated samples in total. We generate 500 and 5,000 samples via ZEROGEN and FEWGEN for comparison. TAMs under all settings are trained three times with different random seeds, and we report the mean and standard deviation in the results. The training process and hyperparameters are detailed in Appendix B.

---

[5] https://openai.com/blog/chatgpt

| Method | #Data | Chinese OntoNotes 4.0 | | | English CoNLL03 | | | Re-TacRED-subset | | | Avg. F1 |
|---|---|---|---|---|---|---|---|---|---|---|---|
| | | P | R | F1 | P | R | F1 | P | R | F1 | |
| PROMPTING | 100 / - | 67.72 | 74.02 | 70.73 | 79.18 | **83.59** | 81.33 | 64.21 | 86.68 | 73.77 | 75.28 |
| SUPERVISED | 100 / - | $70.54_{1.33}$ | $\mathbf{75.66_{1.14}}$ | $73.00_{0.84}$ | $77.16_{0.31}$ | $78.52_{0.52}$ | $77.94_{0.10}$ | $62.36_{2.35}$ | $91.88_{1.90}$ | $74.28_{2.05}$ | 75.07 |
| ZEROGEN | - / 500 | $62.10_{1.70}$ | $71.87_{0.68}$ | $66.62_{1.05}$ | $71.14_{2.64}$ | $71.10_{2.08}$ | $71.07_{0.36}$ | $61.60_{7.21}$ | $78.25_{5.37}$ | $68.57_{3.14}$ | 68.75 |
| | - / 5000 | $62.00_{0.92}$ | $72.84_{2.50}$ | $66.97_{0.61}$ | $74.23_{3.32}$ | $71.78_{1.97}$ | $72.99_{2.61}$ | $51.46_{0.82}$ | $94.28_{0.65}$ | $66.57_{0.66}$ | 68.84 |
| FEWGEN | 100 / 500 | $71.78_{4.34}$ | $71.06_{1.66}$ | $71.35_{1.80}$ | $73.06_{2.31}$ | $69.87_{2.23}$ | $71.43_{2.21}$ | $69.21_{2.49}$ | $77.84_{11.21}$ | $73.12_{6.46}$ | 71.97 |
| | 100 / 5000 | $68.05_{0.81}$ | $75.17_{0.48}$ | $71.43_{0.52}$ | $75.93_{2.67}$ | $72.93_{1.80}$ | $74.40_{2.20}$ | $68.07_{3.08}$ | $92.24_{5.23}$ | $78.20_{0.99}$ | 74.68 |
| LLMAAA-random | 100 / 500 | $68.85_{2.36}$ | $71.63_{2.02}$ | $70.21_{2.00}$ | $77.69_{2.11}$ | $80.75_{1.49}$ | $79.17_{1.32}$ | $63.23_{9.60}$ | $97.75_{2.63}$ | $76.41_{6.48}$ | 75.26 |
| LLMAAA-confidence | 100 / 500 | $\mathbf{72.66_{2.42}}$ | $75.49_{1.67}$ | $\mathbf{74.00_{0.44}}$ | $\mathbf{82.91_{0.83}}$ | $82.78_{0.63}$ | $\mathbf{82.84_{0.31}}$ | $\mathbf{71.49_{4.76}}$ | $93.28_{5.18}$ | $\mathbf{80.79_{2.63}}$ | **79.21** |

Table 1: Evaluation results for LLMAAA and other baselines across three different datasets, using ChatGPT as LLM backbone. We report the mean and standard deviation of three separate runs for each method. Since we set the temperature to 0 in PROMPTING, its results are deterministic and we only run evaluation once. We also denote the amount of data (gold/silver) that TAM used for training.

| Relation | Generated Data |
|---|---|
| per:parents | **Mary**subj's father is **Adam**obj. |
| | **Tom**subj's mother, **Mary**obj, lives in New York. |
| | **Michelle Obama**subj's parents are **Fraser C. Robinson III and Marian Shields Robinson**obj. |
| per:children | **Mike**subj's son is named **Jack**obj. |
| | **Lily**subj's children are **Alex and Bella**obj. |
| | **Sarah**subj has a daughter named **Emily**obj. |

Table 2: A case study of generated data with ZEROGEN on Re-TACRED. We leverage ChatGPT as the text generator, and the full prompts we use can be found in Appendix B.3.

We follow consistent principles in prompt design. Empirically, we find that in-context examples bring marginal benefit to RE, while label verbalizer is a technique specifically designed for the classification task. Therefore, We apply $k$-NN example retrieval to NER and label verbalizer to RE separately. We set $k$ to 5 for all experiments, including FEWGEN. Refer to Appendix B.3 for full prompts.

## 5.2 Overall Results

Table 1 denotes our main experiment results. LLMAAA with least confidence as acquisition function outperforms all comparative baselines across all datasets, with 74.00%, 82.84% and 80.79% F1 scores on Chinese OntoNotes 4.0, English CoNLL03 and Re-TACRED-subset, respectively.

Comparing with PROMPTING (i.e. the LLM annotator), LLMAAA shows steady improvement (4% in average score) with TAMs of much fewer parameters and lower inference latency, indicating that LLMAAA provides a decent substitute for LLMs in real-world deployments. LLMAAA also surpasses SUPERVISED, where TAMs are trained on clean-labeled but smaller-scale data. This suggests that LLMAAA is capable of deriving rich knowledge beyond the limited demonstration/validation set on unlabeled data, which benefits generalization.

We also notice that generation-based methods, i.e. ZEROGEN and FEWGEN, fail to establish on-par results, even with $10\times$ more data in zero-shot setting. We argue that the text-generation abilities of LLMs are exaggerated in complex scenarios. To demystify the illusion, we devise a case study on Re-TACRED, as is shown in Table 2. ZEROGEN tends to generate simple templated sentences that deviate from the news domain, i.e. the original corpus of Re-TACRED. These results may induce low-quality and domain-shift issues that hamper TAMs' performance. FEWGEN's performance improves with in-context examples, however, it still lags far behind LLMAAA. In contrast, exploiting the unlabeled data effectively alleviates these problems with much higher efficiency, where only hundreds of annotated samples are sufficient for satisfactory performance.

## 5.3 Ablations

### 5.3.1 Effects of Prompt Engineering

Though ChatGPT can well follow human instructions in general, it still struggles with difficult tasks and domain-specific data. We compare the infer-

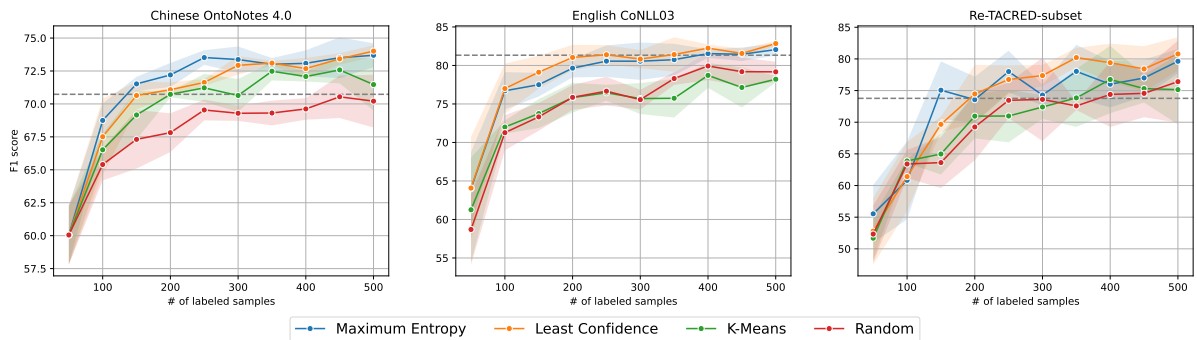

Figure 3: LLMᴀᴀᴀ's performance with different active acquisition strategies, shown by F1 scores. The dashed lines denote Pʀᴏᴍᴘᴛɪɴɢ's results. For each method, we report the mean and standard deviation of three runs initialized with different random seeds.

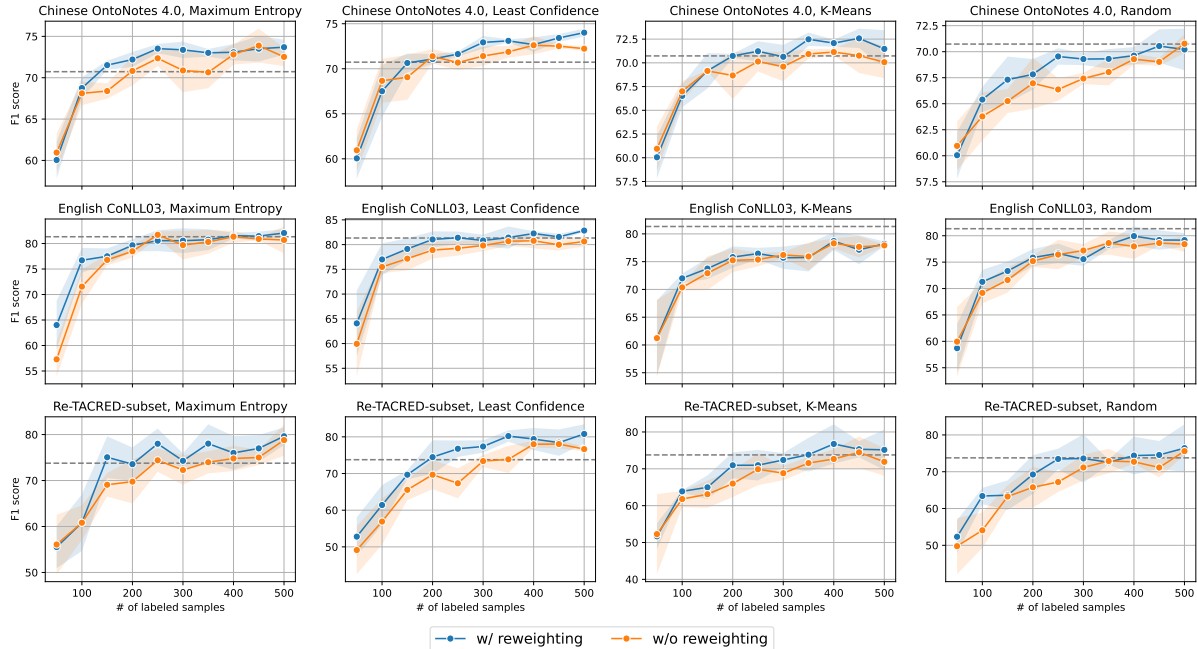

Figure 4: Results for analyzing the effects of automatic reweighting. We remove the online-approximated sample weights and train TAMs with standard loss objectives for ablation. The dashed lines denote Pʀᴏᴍᴘᴛɪɴɢ's performance. For each method, we report the mean and standard deviation of F1 scores within three different runs.

|  | OntoNotes | CoNLL | Re-TacRED |
|---|---|---|---|
| Base Instruction | 49.62 | 55.74 | 70.94 |
| +$k$-NN Examples | **70.73** | **81.33** | - |
| +Label Verbalizer | - | - | **73.77** |

Table 3: Comparison results between plain instructions and optimized prompts in F1 scores.

ence performance of plain instructions with optimized prompts in Table 3. Without $k$-NN example retrieval module (i.e. in zero-shot manners), the LLM annotator is unable to extract entities well in NER task, shown by a drastic drop in F1 scores (21% on OntoNotes and 25% on CoNLL). This result highlights the need for demonstrations, where

LLMs' zero-shot performance is unsatisfactory. In addition, the label verbalizer can help align unnatural labels with natural language descriptions, which improves the performance in RE (from 70.94% to 73.77% in F1). These findings emphasize that prompt engineering is crucial for building strong annotators, and incorporating similar and aligned contexts contributes to better inference.

### 5.3.2 Accelerating with Active Learning

Figure 3 shows LLMᴀᴀᴀ performance with different active learning strategies across all datasets.

Uncertainty-based methods, i.e. **maximal entropy** and **least confidence**, perform significantly better than the random baseline, with faster con-

| Backbone | Method | P | R | F1 |
|---|---|---|---|---|
| GPT-3 | PROMPTING | 41.82 | 22.77 | 29.49 |
| | LLMAAA-confidence | **57.26** | **56.09** | **56.63** |
| ChatGPT | PROMPTING | 67.72 | 74.02 | 70.73 |
| | LLMAAA-confidence | **72.66** | **75.49** | **74.00** |
| GPT-4 | PROMPTING | 68.70 | **79.42** | 73.68 |
| | LLMAAA-confidence | **73.47** | 76.42 | **74.90** |

Table 4: Results on Chinese OntoNotes 4.0 for PROMPTING and LLMAAA with different LLMs. LLMAAA uses least confidence as the acquisition function, and annotates 500 samples for TAM training.

vergence and higher F1 scores at the end of iterations. It is worth noting that (1) uncertainty-based methods are able to achieve on-par performance with random selection with only 30%~40% training data, (2) they surpass PROMPTING consistently within 500 LLM-annotated training samples. In summary, uncertainty-based active learning strategies enable LLMAAA to be more efficient and more capable.

Though $k$-**means** clustering encourages diversity in feature space, it only outperforms random sampling on OntoNotes, while yielding similar results on CoNLL03 and Re-TacRED. This suggests that it may require more training data for finetuned BERT to learn informative representations, and such a diversity-based method may fail in low-resource environments, e.g. at early iterations of the loop.

### 5.3.3 Reweighting Helps Robust Training

Figure 4 depicts the learning trials with and without the automatic reweighting technique. We observe that reweighting training samples consistently help improve performance across all datasets and methods. This finding proves that the training process of TAMs is more noise-tolerant with automatic reweighting, even with only a small-scale clean-labeled set (100 samples) serving for validation.

In particular, the performance gain from automatic reweighting is more prominent on OntoNotes and Re-TACRED, and diminishes on CoNLL03. We argue that automatic reweighting plays a crucial role when the LLM annotators are relatively poor (as in OntoNotes and Re-TACRED). In such scenarios, the online approximation of the validation set serves as an effective estimation of unbiased data distribution, and helps prevent TAMs from overfitting noisy labels.

## 6 Analysis

### 6.1 LLMAAA with Different Annotators

To guarantee the universal effectiveness of LLMAAA, we further investigate the performance with other LLM annotators, i.e. GPT-3 (Brown et al., 2020) and GPT-4 (OpenAI, 2023). Due to budgetary considerations, we opt to restrict our experiments to OntoNotes. The precision, recall and F1 score are shown in Table 4. The results indicate that LLMAAA benefits from better annotators with continuous improvements, and more importantly, TAMs trained by LLMAAA outperform the LLM annotators consistently. The student outperforms the weak teacher by a large margin (27% in F1 for GPT-3). As the teacher grows stronger, this gap narrows down. This trend meets our expectations: since student TAMs are trained with a fixed budget of data (500 samples), enhancing the capabilities of teacher LLMs will gradually approach the performance ceiling of the students. More annotation budget and more powerful TAMs can help extend this limit, while we leave the exploration for future research.

### 6.2 Why Can Students Outperform Teachers?

An interesting observation across our experiments is that student TAMs trained with generated labels can outperform teacher LLMs, i.e. LLMAAA > PROMPTING, even without sample reweighting, as shown by Figure 4. Such results partially align with previous findings in knowledge distillation (Wang, 2021; Song et al., 2021) and pseudo-label-based learning (Lee, 2013; Sanyal et al., 2022; Min et al., 2023), which share similar yet slightly different settings with LLMAAA.

We attempt to further explain the phenomenon in a simplified setting, where we consider a binary classification task that predicts $y$ for $\mathbf{x} \sim \mathcal{D}(\mathbf{x})$, where $\mathcal{D}(\mathbf{x})$ is discrete as in language space. For simplicity, we let $y = 1$ denote the **correct** label and $y = 0$ otherwise. We first make the natural assumption that the teacher's performance is above chance, i.e. the accuracy $p > 0.5$. Querying teacher for target sample $\mathbf{x}_t$ will generate pseudo label $y_t \sim \text{Bernoulli}(p)$. If the student is a universal function approximator $S(\mathbf{x}; \theta)$ that outputs a scalar as probability that $\hat{y} = 1$, then minimizing the

cross-entropy loss

$$\min_{\theta} \mathbb{E}_{\mathbf{x}_t \sim \mathcal{D}(\mathbf{x}), y_t \sim \mathcal{B}(p)}[-y_t \log(S(\mathbf{x}_t; \theta))$$
$$-(1 - y_t) \log(1 - S(\mathbf{x}_t; \theta))]$$

will reach optimal with $S(\mathbf{x}; \theta) = p$. Usually we predict with heuristics that $\hat{y} = 1$ if $S(\mathbf{x}; \theta) > 0.5$. With the previous assumption, we have $\hat{y} = 1$, which means that $S$ always predicts correctly. This toy case nonetheless explains that *an ordinary teacher can raise better students*. Though teacher LLMs are deterministic for specific $\mathbf{x}$ when the temperature is set to 0, their predictions are yet statistically random in $\mathcal{D}(\mathbf{x})$, where the same conclusion holds.

We shall point out that the above discussion considers a much-relaxed setting, where we attempt to account for an intuitive understanding on why students outperform teachers in the hard label distillation problem. We leave the rigorous theoretical analysis for future work.

## 7 Conclusion

In this work, we propose LLMAAA, a framework that uses LLMs as active annotators to address the challenges of data scarcity in NLP tasks. With active learning strategies, LLMAAA allows LLMs to label more informative samples that promote TAMs performance efficiently. We also optimize for reliability within the framework, which uses prompt engineering techniques and automatic reweighting to improve annotation quality and to reduce the impact of noisy labels, respectively. Experiments on NER and RE tasks demonstrate the effectiveness of LLMAAA. The evaluation results highlight the **efficiency** and **reliability** of LLMAAA. Trained with just hundreds of LLM-annotated samples, TAMs are able to outperform their teacher LLMs substantially. Besides, LLMAAA is also much more efficient compared to prevalent data generation methods, which usually require orders of magnitude more synthetic training data. These findings reveal that LLMAAA offers a cost-effective, privacy-ensured, yet well-performing solution to apply LLMs in practical scenarios.

## Limitations

Although LLMAAA demonstrates success in transferring and exceeding LLMs' capabilities with cheaper TAMs, it does come with certain limitations. The main difference between the setting in LLMAAA and previous zero-shot generation-based methods, e.g. ZEROGEN and SUNGEN, is that we use an unlabeled data pool $\mathcal{D}_{\text{pool}}$ and oracle-annotated data $\mathcal{D}_{\text{demo}}/\mathcal{D}_{\text{val}}$, to provide extra knowledge. However, we shall point out that unlabeled text is readily available in many real-world scenarios, thus it is practical to make the pool-based assumption. Additionally, in complex tasks where zero-shot inference fails (like NER in our experiments), it is costly yet necessary to incorporate demonstrations for LLMs. In LLMAAA, we strive for minimizing human efforts by restricting the oracle-annotated data to a small scale (100 samples), and exploiting the same data for demonstration and validation. Another bottleneck is the model capacities of teacher LLMs and student TAMs. On one hand, a weak teacher is unable to teach excellent students that are ready to be used for applications (e.g. GPT-3). On the other hand, TAMs are bounded depending on their architectures. When the teacher surpasses the ceiling, it will be theoretically impossible for students to outperform teachers. Despite these cases, we are optimistic that LLMAAA is effective in most situations.

We adopt the proprietary GPT family as annotators in experiments, which are provided by OpenAI in a black-box manner. Though powerful, this practice may raise several concerns, e.g. the potential exposure to test data. Nevertheless, we believe that given the comprehensive analysis in § 6.1, it does not affect the effectiveness of our method.

## Ethics Statement

This work utilizes publicly available benchmark datasets, and we respect and adhere to their licenses and agreements. Our proposed method involves the use of LLMs for data annotation, as discussed in GPT3Mix (Yoo et al., 2021). This paradigm still poses several challenges, such as the potential biases or toxic content in the generated data. Therefore, it is crucial to exercise caution when employing our method to invoke LLMs for generating data and when utilizing TAMs trained on such generated data. Applying our work to downstream tasks such as NER and RE may result in issues such as mis-extraction and false information, and may fail in some cases. When employing our method, it is essential to consider using debiasing (Schick et al., 2021) or manual checking to mitigate these concerns.

## Acknowledgements

We appreciate the anonymous reviewers for their valuable advice on this manuscript. We would like to thank Tianhao Wu for the insightful discussion and feedback. This work was supported by NSFC under grant 61932001 and U20A20174. The corresponding author of this paper is Lei Zou (zoulei@pku.edu.cn).

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

# A  Dataset Statistics

In this section, we describe the statistics and pre-processing of each dataset in detail.

**OntoNotes 4.0** (Weischedel et al., 2011) is a large corpus comprising various genres of text (news, web text, etc.) in three languages (English, Chinese, and Arabic) with structural information and shallow semantics, and has been widely used for NER. We use the Chinese text data and take the same data split as Che et al. (2013), which uses four most common entity types, i.e. PER (person), LOC (location), ORG (organization) and GPE (geo-political entities). We truncate token length within 512 (i.e. split long input to multiple chunks) to

| | Training | Validation | Testing |
|---|---|---|---|
| per:age | 421 | 256 | 208 |
| per:nationality | 295 | 222 | 115 |
| per:parents | 182 | 69 | 106 |
| per:children | 275 | 114 | 55 |
| per:siblings | 211 | 33 | 66 |
| per:spouse | 271 | 189 | 73 |
| no_relation | 2,482 | 1,324 | 934 |
| **Total** | 4,137 | 2,207 | 1,557 |

Table 5: Statistics of relation types in each split of Re-TACRED-subset. We replace the label "per:origin" with "per:nationality" for clarity.

fit in BERT input limit. The processed data contains 15,724/4,301/4,346 samples for training/validation/testing, respectively.

The **English CoNLL 2003** shared task (Tjong Kim Sang and De Meulder, 2003) is a NER dataset that contains four entity types: PER (person), LOC (location), ORG (organization), and MISC (miscellaneous entities), which consists of 14,041/3,250/3,453 sentences for training/validation/testing.

**Re-TACRED** (Stoica et al., 2021) is a revised version of TACRED (Zhang et al., 2017), a large-scale crowdsource-annotated RE dataset. It originally has 40 relation types. Including all these types will lead to much longer prompts, which may exceed the API length limit and receive responses with higher latency. Therefore, we opt to select a subset of relations that describe personal relationships for study. We keep all these relation instances in training/validation/testing sets, and balance the NA relation instances to the original portion. The statistics for each relation type is shown in Table 5.

For all three datasets, we randomly sample 100 examples from the original validation sets and reuse the same data for demonstration $\mathcal{D}_{\text{demo}}$ and validation $\mathcal{D}_{\text{val}}$. We use the full training sets as the initial $\mathcal{D}_{\text{pool}}$, from which we randomly sample active learning's seed labeled sets $\mathcal{D}_{\text{labeled}}$ with a size of 50.

## B Implementations

### B.1 LLM Inference APIs

We access OpenAI APIs by Azure service. The API we use for each model is depicted in Table 6. Since ChatGPT and GPT-4 will continue to be updated, they may generate different responses as time changes, even when the temperature is 0.

| Model | API |
|---|---|
| GPT-3 | text-curie-001 |
| ChatGPT | gpt-35-turbo |
| GPT-4 | gpt-4 |

Table 6: Azure OpenAI service API that we use.

### B.2 Training Task-Specific Models

For all experiments that train TAMs for inference (i.e. LLMAAA, ZEROGEN, FEWGEN and SUPERVISED), we repeat each with three random seeds, resulting in different parameter initialization and random data sampling. We report the mean and standard deviation in our results.

We use bert-base-cased (Devlin et al., 2019) as TAMs' encoders with a learning rate of 5e-5 for English data (CoNLL03 and Re-TACRED), and chinese-bert-base-wwm (Cui et al., 2021) with a learning rate of 2e-5 for Chinese data (OntoNotes 4.0). The learning rate of other parameters (i.e. linear classifiers) is set to 1e-4. We optimize the models via AdamW (Loshchilov and Hutter, 2019), with $\epsilon = $ 1e-6, under a linear warmup schedule for the first 6% steps. We train all TAMs with a batch size of 8 for 40 epochs and take the checkpoint with the highest validation performance for final prediction.

### B.3 Prompts

The full prompts we use for annotation are shown in Table 7. In Re-TACRED, we provide prompts both with and without verbalized labels. To add demonstration, we insert each sample's text into input and label to output. The target sample is added to the last input, and the last output is left blank for prediction.

We also show the prompts for generation in Table 8. We use them similarly to annotation. In the zero-shot setting, to help models generate desired outputs, we use a default example to inform LLMs about the output format.

## C Annotation Examples

We show two annotation examples of correct/partially wrong annotations from the CoNLL 2003 NER dataset in Listing 1. The first example is exactly correct, and the second example contains hallucinations that do not exist in ground truth: "April", "March", and "Thursday".

| Task | Prompting | |
|---|---|---|
| CoNLL 03 | **Description** | You are a highly intelligent and accurate news domain named-entity recognition (NER) system. You take passage as input and your task is to recognize and extract specific types of named entities in that given passage and classify into a set of following predefined entity types: [person (PER), location (LOC), organization (ORG), miscellaneous entity (MISC)] Your output format must be in json form of: ["span": span, "type": type, ...] |
| | **Instruction** | The span must be exactly the same as the original text, including white spaces. |
| | **Format** | **Input**: "Input: {}" 
 **Output**: "Output: {}" |
| OntoNotes 4.0 | **Description** | 你是一名通用领域的命名实体识别（NER）标注者，给定一段输入文本和 NER 类型，你需要以 json 格式返回 NER 的 span 和类型。
 类型：[人物（PER），组织机构（ORG），地缘政治实体（GPE），地理位置（LOC）]
 输出格式：["span": span, "type": type, ...] |
| | **Format** | **Input**: "输入：{}" 
 **Output**: "输出：{}" |
| Re-TACRED (Original) | **Description** | Given a sentence, and two entities within the sentence, classify the relationship between the two entities based on the provided sentence. If no relation of interest exists, strictly return "no_relation". All possible relationships are listed below: |
| | **Instruction** | - per:age 
 - per:parents 
 - per:spouse 
 - per:siblings 
 - per:children 
 - per:nationality 
 - no_relation |
| | **Format** | **Input**: "Sentence: {}" 
 **Output**: "Relationship: {}" 
 **Struct**: "e1: {} 
       e2: {}" |
| Re-TACRED (Verbalized) | **Description** | Given a sentence, and two entities within the sentence, classify the relationship between the two entities based on the provided sentence. If no relation of interest exists, strictly return "no_relation". All possible relationships and explanations are listed below: |
| | **Instruction** | - per:age : the age of {e1} is {e2} 
 - per:parents : {e1}'s parent is {e2} 
 - per:spouse : {e1}'s spouse is {e2} 
 - per:siblings : {e1} is the sibling of {e2} 
 - per:children : {e1}'s children is {e2} 
 - per:nationality: {e1}'s nationality is {e2} 
 - no_relation : {e1} has no known relations to {e2} |
| | **Format** | **Input**: "Sentence: {}" 
 **Output**: "Relationship: {}" 
 **Struct**: "e1: {} 
       e2: {}" |

Table 7: Annotator's prompts for each task.

```
{
    "text":"Celtic 's Jackie McNamara , who did well with last season 's successful
        under-21 team , earns a call-up to the senior squad .",
    "labels":[{"span":"Celtic","type":"ORG"},{"span":"Jackie McNamara","type":"PER
        "}]
},
{
    "text":"Finland 's trade surplus rose to 3.83 billion markka in April from 3.43
        billion in March , the National Customs Board ( NCB ) said in a statement on
         Thursday .",
    "labels":[{"span":"NCB","type":"ORG"},{"span":"Finland","type":"LOC"},{"span":"
        National Customs Board","type":"ORG"},{"span":"April","type":"MISC"},{"span
        ":"March","type":"MISC"},{"span":"Thursday","type":"MISC"}]
}
```

Listing 1: Annotation examples.

| Task | Prompting | | |
|------|-----------|---|---|
| CoNLL 03 | **Description** | You are an intelligent text data generator. Generate {} high-quality and diverse sentences in news domain containing entities for the following types:
[person (PER), location (LOC), organization (ORG), miscellaneous entity (MISC)]
Write one sample per line. No other output. | |
| | **Format** | **Example**: "Example: {}"
**Output**: "Output: {}" | |
| | **Default Example** | {"text": text, "entities": [{"name": name, "type": type}]} | |
| OntoNotes 4.0 | **Description** | 你是一名新闻领域的文本生成助手。生成{}个流畅、通顺、多样的中文句子，并包含下面这些类型的命名实体（entity）：
[人名（PER），组织机构名（ORG），地缘政治实体（GPE），地理位置（LOC）]
每行输出一个样本，不要有任何额外的输出。 | |
| | **Format** | **Example**: "示例：{}"
**Output**: "输出：{}" | |
| | **Default Example** | {"text": text, "entities": [{"name": name, "type": type}]} | |
| Re-TACRED | **Description** | You are an intelligent text data generator. Generate {} high-quality and diverse sentences in news domain containing relational triplet for the following relation types:
- per:age : the age of SUBJ is OBJ
- per:parents : SUBJ's parent is OBJ
- per:spouse : SUBJ's spouse is OBJ
- per:siblings : SUBJ is the sibling of OBJ
- per:children : SUBJ's children is OBJ
- per:nationality: SUBJ's nationality is OBJ
- no_relation : SUBJ has no known relations to OBJ
Write one sample per line in json format. Subject and object must appear in the sentence. No other output. | |
| | **Format** | **Example**: "Example: {}"
**Output**: "Output: {}" | |
| | **Default Example** | {"text": text, "subject": subject, "object": object, "relation": relation} | |

Table 8: Generator's prompts for each task.