# OpenReview forum: "LLMaAA: Making Large Language Models as Active Annotators"
_EMNLP/2023/Conference — EMNLP 2023 Findings_

### Official Review · Reviewer_z8rH · 2023-08-03

**Soundness:** 3

**Excitement:**

4: Strong: This paper deepens the understanding of some phenomenon or lowers the barriers to an existing research direction.

**Paper Topic And Main Contributions:**

The paper is about building information extraction models (evaluations include named entity recognition and relation labeling) using language models.  The key idea is to augment a small annotated dataset with unlabeled instances plus labels produced by a "large" language model like ChatGPT; the combined gold and silver datasets are then used to finetune a cheaper model (here, BERT) to perform the IE task.  The experimental results show that on three datasets this approach is better than merely prompting the large model or using only the gold data to train the smaller model.

**Questions For The Authors:**

Question 1:  No justification is given for the selection of the BERT student model.  Is it a strong starting point known to achieve near state-of-the-art performance?  Was it chosen because it satisfied some budget constraints?

**Reasons To Accept:**

The technical contribution of this work is essentially to assemble a collection of established techniques into a larger pipeline to experiment with distillation in a setting where I don't think it's been explored before.  Because the components are mostly (with some exceptions, see below) reasonably well motivated and explained, this is fine.  The experimental results show that, at least under the explored settings, one can improve the "student" model using noisily annotated data from the large LM teacher.

I did appreciate the care the authors took in section 6.1, where different teachers are considered.  The approach does consistently outperform basic prompting across several strong models, though the gap is closing as we get to GPT-4 (not discussed).

**Reasons To Reject:**

The paper misuses the term "active learning."  Active learning uses a trained model to select examples for annotation, but employs humans to annotate them.  The method here takes the trained model's labels as given, without having any humans in the loop.  If these were being fed back into the same model iteratively, we'd have some version of bootstrapping or self-training.  As it is, they are being used to train a secondary model, making this effectively a kind of distillation.  (The paper even adopts distillation terminology -- "teacher" and "student.")

There is no discussion of the source of the unlabeled data used in the experiments.  This seems like it could have a huge effect on the methods.

The paper makes claims about efficiency and privacy that are not well supported.  The privacy question could potentially be addressed simply by making the argument explicit; it seems to be along the lines of "if you do supervised training of your own model, you don't have to upload your data to the LLM service provider."  But in fact with the method presented in this paper, the labeled and unlabeled data *do* need to be uploaded, to prompt the LLM to generate labels.

As for efficiency, the paper could do a much better job of laying out the costs of the range of approaches it considers.  There's the cost of having humans annotate gold data, the cost of the LM service to do the generations, and the computational cost of finetuning BERT.  None of these are actually quantified.  The main experiments of the paper do present details about how much data of each kind each method uses, but they don't explore how the performance varies as a function of any of these.  Ultimately we have arbitrary choices about how much gold data and how much of each kind of silver data are used for each approach.  The experiments might have turned out very differently if the authors had started with more gold data, for example.  Or perhaps if we used a more powerful or better pre-trained student model, we could get away with less OpenAI-based annotation.  The paper would be much strongly if there was more sensitivity analysis to these choices, paying particular attention to providing useful guidance about the tradeoffs among these various costs.

The use of proprietary language models as a component is a limitation that should at the very least be discussed.  In particular, the authors and reviewers have no way of knowing whether ChatGPT or the other large models were exposed to the test data during their construction.

**Reproducibility:**

2: Would be hard pressed to reproduce the results. The contribution depends on data that are simply not available outside the author's institution or consortium; not enough details are provided.

**Reviewer Confidence:**

4: Quite sure. I tried to check the important points carefully. It's unlikely, though conceivable, that I missed something that should affect my ratings.

**Typos Grammar Style And Presentation Improvements:**

The paper would be improved with careful editing.  From the very beginning, there's a lot of repetition (e.g., "Prevalent supervised learning methods in NLP are notoriously data-hungry, which demand large amounts of high-quality annotated data.  In practice, acquiring such data is a costly endeavor." could be as simple as "This paper considers the cost of high-quality annotated data required by current supervised learning methods in NLP.")  The opening discussion about language models as a service is pretty disconnected from the contributions of the paper.  I'd recommend revising the intro to focus directly on the paper's contributions.

Line 090, I don't understand what this means:  "building fine-level descriptions aligned with natural language for uninformed labels"

Line 125, the phrase "real world miniaturization" is misleading; this is not a paper about real-world applications or miniaturization

Figure 2:  the numbered elements in the caption don't have analogues in the figure itself

The discussion of the label verbalizer (line 233) is not self-contained, essentially assuming the reader is familiar with all the details of past work.

---

> ### Author Rebuttal · Authors · 2023-08-28
>
> Thanks for your detailed and constructive comments, and they are exceedingly helpful to improve our paper. We sincerely appreciate your time in reading the paper, and our point-to-point responses to your comments and questions are given below.
>
> ### Comments:
>
> > C1: The paper misuses the term "active learning".
>
> The term itself emphasizes the label acquisition process from an “oracle”, and human annotator is just one specific instance ([Settles, 2009](https://burrsettles.com/pub/settles.activelearning.pdf)). Therefore, we think that this term properly describes our framework, if we take LLM as the oracle in the loop. Nevertheless, thanks for your suggestions, and we realize that it will bring some misunderstanding. We will revise our usage of “active learning” in our writing and emphasize “active acquisition” more instead.
>
> > C2: There is no discussion of the source of the unlabeled data used in the experiments. This seems like it could have a huge effect on the methods.
>
> In our experiments, the unlabeled data pool is the original training sets of `zh_ontonotes`/`en_conll03`/`re-tacred-subset`, and we do not use their ground truth labels throughout the experiments. The gold data for demonstration/validation is randomly sampled from the original validation sets, and we conduct evaluation on the original test sets. So the unlabeled data and gold data comes from the same domain, as the original training and validation sets. We will clarify the setting in our implementation section.
>
> > C3: Question on privacy: with the method presented in this paper, the labeled and unlabeled data do need to be uploaded, to prompt the LLM to generate labels.
>
> We mainly aim to tackle the online privacy and costs issues. Using a offline finetuned model for inference avert these problems. As for the training stage, user can control what data to upload (by restricting the labeled and unlabeled sets) and use data masking techniques to further alleviate the privacy issue.
>
> > C4: Question on efficiency: the main experiments of the paper do present details about how much data of each kind each method uses, but they don't explore how the performance varies as a function of any of these (gold data, silver data, and student model). The experiments might have turned out very differently if the authors had started with more gold data, for example. Or perhaps if we used a more powerful or better pre-trained student model, we could get away with less OpenAI-based annotation. The paper would be much stronger if there was more sensitivity analysis to these choices, paying particular attention to providing useful guidance about the tradeoffs among these various costs.
>
> We totally understand your concerns. We first leave alone the computation cost of finetuning (which is relatively small compared to the other two), and analyze #gold and #silver. Obviously, the performance is an monotonic increasing function of #gold and #silver converging to the ceiling, so the problem here reduces to discuss the tradeoff between #gold and #silver, given a fixed budget.
> However, we are afraid that the ratio #gold/#silver may vary among different tasks and even different budgets, therefore providing a grid search of the Pareto frontier for a specific budget is meaningless.
>
> Here we try to provide another line of analysis to this problem.
> - We first estimate the cost of gold and silver annotation for each sample. According to ([Wang et al., 2021](aclanthology.org)), the cost of human annotation is roughly estimated as \\$2/1k tokens, while ChatGPT charges for \\$0.002/1k tokens. Even taking the additional cost of template into account, we believe that human labeler can be 10x to 100x more expensive than LLM. So \\$gold is much higher than \\$silver, and should be optimized in priority.
> - Therefore, we can decouple the process by minimizing #gold first, then #silver.
> We then propose a design principle of choosing #gold and #silver, given an expected performance $p$. Recall that Table 4 shows that the student performance is strongly correlated to its teacher with a gap, so we can set a threshold $\tau$ and grid search #gold to let LLM reach $p - \tau$, with a few test data for estimation. We can then assign remaining budget to #silver, or early stops until the final performance saturates. Since LLMaAA asks for silver data is in a iterative loop fashion, the early stopping thus can be performed easily.
>
> We will add this part of discussion into the analysis section or the appendix if you find it helpful. If you have any other suggestions or concerns on this issue, please do not hesitate to leave your further comments.
>
> > C5: The use of proprietary language models as a component is a limitation that should at the very least be discussed. In particular, the authors and reviewers have no way of knowing whether ChatGPT or the other large models were exposed to the test data during their construction.
>
> Thanks for the suggestion, and we will add a discussion on the use of proprietary language models in the limitation section.
>
> ### Questions:
>
> > Q1: No justification is given for the selection of the BERT student model. Is it a strong starting point known to achieve near state-of-the-art performance? Was it chosen because it satisfied some budget constraints?
>
> We want to incorporate least inductive bias in the choice of student model, so we build two simple models with BERT as encoders, which is just a common baseline, not sota. We want to show that LLMaAA works fine with such simple student models, without the need of “heavy” design heuristics or strong encoders.
>
> ### Typos Grammar Style And Presentation Improvements:
>
> Thanks for your instructive suggestions and comments. We will carefully refine our abstract, introduction and writing style and modify Figure 2’s caption accordingly. We will also include a brief example of label verbalizer in our revised version. As for now, you can refer to Table 7 in Appendix C to see how verbalizer works.

---

### Official Review · Reviewer_P3tf · 2023-08-04

**Soundness:** 3

**Excitement:**

4: Strong: This paper deepens the understanding of some phenomenon or lowers the barriers to an existing research direction.

**Missing References:**

N/A

**Paper Topic And Main Contributions:**

The authors proposed a method of how samples should be annotated efficiently using large language models for sequence labeling tasks. To effectively leverage pseudo-labels, the authors first draw k-NN samples and adopt the automatic reweighting technique to assign training samples with learnable weights. They successfully incorporate LLMs-generated noisy data and achieve competitive performance on NER and RE tasks compared to strong baselines.


**Questions For The Authors:**

Q1. While the authors evaluate their method on low-resource settings (which is reasonable) just curious about how will the performance change when increasing LLMaAA generated data. In other words, if using 5,000 data instead of 500, is the performance still competitive compared to other baselines?

Q2. What do the actual examples look like that are generated by the proposed method? As shown in Table 2, it would be interesting to see the case study of what actual samples generated and annotated by LLMaAA look like.


**Reasons To Accept:**

The paper is easy-to-follow and well-organized.

**Reasons To Reject:**

Is there any chance to explore how LLMaAA will work on different numbers of labeled sample settings? The authors only provide a single setting which uses 500 samples to evaluate their method. Since their baseline methods FEWGEN and ZEROGEN show performance degradation when using larger amounts of data, it is a concern that LLMaAA would show similar trends. It would be interesting to see how the proposed method will work in a very low resource setting (10 samples) or in a large number of labeled data.

**Reproducibility:**

3: Could reproduce the results with some difficulty. The settings of parameters are underspecified or subjectively determined; the training/evaluation data are not widely available.

**Reviewer Confidence:**

4: Quite sure. I tried to check the important points carefully. It's unlikely, though conceivable, that I missed something that should affect my ratings.

**Typos Grammar Style And Presentation Improvements:**

Figure 4 shows can be simplified by providing the subsets of graphs. Since it is cluttered to show all graphs on different automatic reweighting methods on all datasets, it would be good to show a single dataset on all automatic reweighting methods, and the remaining can be moved to a supplementary section.

---

> ### Author Rebuttal · Authors · 2023-08-28
>
> Thanks for your positive feedback and valuable comments. We sincerely appreciate your time in reading the paper, and our point-to-point responses to your comments and questions are given below.
>
> ### Comments & Questions:
>
> > C1 & Q1:  How the proposed method will work in a very low resource setting (10 samples) or in a large number of labeled data.
>
> As a brief recap, we use 100 gold data (ground truth) for demonstration and validation, and we use 500 silver data (LLM-annotated) to train small models. According to the context, we assume that here you ask about the number of silver data.
> - Under very low resource setting (i.e. 10 samples), training small models even with gold data still can’t achieve satisfactory generalization performance, therefore it seems to be trivial to discuss such a small scale.
> - With a larger number of data, a more suitable analogy of LLMaAA should be SUPERVISED (i.e. training with gold data) rather than ZeroGen/FewGen, which are broadly discussed in active learning literature: the performance keeps increasing until saturation. This trend also meets our results in Figure 3. We believe that with a large number of silver data, LLMaAA performance can further improve (but not so much). Besides, in Table 1, the only performance degradation is ZeroGen on Re-TacRED, so we also concern that the degradation trend does not generally hold.
>
> > Q2:  Case study of LLMaAA annotations.
>
> Sure, here is two examples of correct/partially wrong annotations on `en_conll03` (NER) task:
>
> ```json
> {"text":"Celtic 's Jackie McNamara , who did well with last season 's successful under-21 team , earns a call-up to the senior squad .","labels":[{"span":"Celtic","type":"ORG"},{"span":"Jackie McNamara","type":"PER"}]}
> {"text":"Finland 's trade surplus rose to 3.83 billion markka in April from 3.43 billion in March , the National Customs Board ( NCB ) said in a statement on Thursday .","labels":[{"span":"NCB","type":"ORG"},{"span":"Finland","type":"LOC"},{"span":"National Customs Board","type":"ORG"},{"span":"April","type":"MISC"},{"span":"March","type":"MISC"},{"span":"Thursday","type":"MISC"}]}
> ```
> The first example is exactly correct, and the second example contains hallucinations that does not contained in ground truth: `{"span":"April","type":"MISC"}`, `{"span":"March","type":"MISC"}` and `{"span":"Thursday","type":"MISC"}`.
>
> ### Typos Grammar Style And Presentation Improvements:
>
> Thanks for your suggestions. We will modify Figure 4 if there is not enough space in later revision.
>
> At the very last, LLMaAA targets at general NLP tasks beyond sequence labeling (i.e. RE is a text classification task). We hope that our responses help address your concerns.

---

### Official Review · Reviewer_Neog · 2023-08-05

**Soundness:** 3

**Excitement:**

3: Ambivalent: It has merits (e.g., it reports state-of-the-art results, the idea is nice), but there are key weaknesses (e.g., it describes incremental work), and it can significantly benefit from another round of revision. However, I won't object to accepting it if my co-reviewers champion it.

**Missing References:**

- The claim in lines 62-64 should be justified with a reference.
- This is a contemporary paper that is related:
Bansal, P., & Sharma, A. (2023). Large Language Models as Annotators: Enhancing Generalization of NLP Models at Minimal Cost. ArXiv, abs/2306.15766.
- This paper may be worth citing to provide additional evidence of students surpassing their teachers:
[Distilling Step-by-Step! Outperforming Larger Language Models with Less Training Data and Smaller Model Sizes](https://aclanthology.org/2023.findings-acl.507) (Hsieh et al., Findings 2023)

**Paper Topic And Main Contributions:**

This paper is on the topics of LLMs, automatic data labeling, and active learning.

The primary contribution of this paper is LLMaAA, an active learning framework for using LLMs to generate labels for downstream models.
The general idea is to replace the human labeler in the standard active learning setup with an LLM.
LLM labels are generated using in-context learning, and in-context examples are selected from a small pool of labeled examples using kNN search.
The acquisition functions used to select instances for labeling are based on the outputs of the downstream model, and are all taken from prior work.
To validate the effectiveness of this approach, the paper also contributes experimental results for LLMaAA on named entity recognition and relation extraction.
The results demonstrate that LLMaAA achieves superior performance to prompting and data synthesis methods, and an ablation study also characterizes the relative effectiveness of different acquisition functions and other design decisions.

**Questions For The Authors:**

a. I couldn't quite follow the description of how label verbalization is performed. Were label verbalizers from the cited works used or did you use a method from one of the cited works to create your own verbalizers?

b. Are the labels provided by the LLM ever used in the set of examples drawn from for in-context learning?

c. In Table 1, the amount of data used for the PROMPTING baseline is '-', however, don't the prompts include in-context learning examples? Is this just the amount of data used by the downstram model? If so, I think it would be helpful to include the amount of data used by the LLM as well.

d. Similarly how many in-context examples are used for FewGen?

**Reasons To Accept:**

- LLMaAA successfully outperforms other learning methods in low- to medium-data settings.
  These results are valuable to the research community for two reasons: 1. LLMaAA provides a successful illustration of how to combine active learning with LLMs, and 2. the results also show the relative ineffectiveness of data synthesis and few-shot prompting.
- The ablation results appear to be fairly comprehensive and provide useful perspective on the relative importance of: teacher strength, different acquisition functions, example reweighting, etc.

**Reasons To Reject:**

- While I understand that language models have improved considerably in the past few months, and that this has created a corresponding need for research into how these models can be used to improve models that are more efficient/cost-effective, I am not sure that I would consider the approach in this paper novel.
  Embedding-based retrieval for in-context learning, using LLMs to generate training data, and the retrieval functions used in the active learning component of this paper all stem from prior work.
  Although I do not consider combining these things trivial, this does make me less excited about the work.
I also have some reservations about the setting studied in this paper:
- Some of the design decisions like the number of initial ground truth examples, and the number of labeled examples seem arbitrary.
  Given that cost-effectiveness is a primary motivation, it would have been more satisfying to see what the performance of, e.g., LLaMAA vs. finetuning if finetuning was performed on a number of samples equivalent in cost to the additional samples from the LLM.
  If the authors could provide some justification for these decisions I would consider increasing my soundness score.
- I am not sure I understand the motivation/value behind the "medium-data" setting studied in this paper.
  The datasets contain sufficiently many (hundreds to thousands of) examples not to be few-shot, but at the same time they are not big enough / sufficiently high quality that the downstream classifier's performance is near fully supervised BERT model.
  This negatively impacted my excitement score.
- I didn't find the argument in Section 6.2 particularly convincing.
  Sure if a model is always more likely to be right than wrong, and you can draw infinitely many labels from it for a given test instance, then a "student" model that returns the most likely label is always right.
  This is an extremely idealized setting that is incongruent with the one studied in the paper:
        a. the whole motivation of active learning in this paper is that drawing labels from teachers is expensive; in cases where you only have finite labels there is a non-zero probability the student will be wrong,
        b. the assumption the teacher is always more likely to be right on every instance is, in my opinion, very unnatural, and
        c. if I understand correctly, the claim that s->p only holds on examples that are being labeled by the teacher, it has no bearing on the student's performance on unseen instances at test time.
  This section had a minor negative impact on my soundness score for the paper.

**Reproducibility:**

3: Could reproduce the results with some difficulty. The settings of parameters are underspecified or subjectively determined; the training/evaluation data are not widely available.

**Reviewer Confidence:**

4: Quite sure. I tried to check the important points carefully. It's unlikely, though conceivable, that I missed something that should affect my ratings.

**Typos Grammar Style And Presentation Improvements:**

- In the abstract I'd recommend replacing "the teacher" with "its teacher LLM" for clarity.
- Figure 1 seems to imply your method does not involve in-context learning and prompting LLMs, as these boxes are omitted from 1.c, however these things are actually crucial to the method.
  Furthermore, the figure also implies that the method does not require labeled data, however a seed pool of 100 examples.
- Efficiency and reliability---the primary purported benefits of the proposed method---are never formally defined.
  While I could infer what is meant by these terms from the rest of the paper, (reliability = accuracy, efficiency = data efficiency), I feel it would be helpful to make these terms more clear.
- In my opinion, some of the language in this paper is a bit sensational, e.g., the word 'deified' in line 404, and 'arcane' in 430.
- 590 well effective

---

> ### Author Rebuttal · Authors · 2023-08-28
>
> Thanks for your constructive and insightful comments, and they are exceedingly helpful for us to improve our paper. We sincerely appreciate your time in reading the paper, and our point-to-point responses to your comments and questions are given below.
>
> ### Comments:
>
> > C1: Some of the design decisions like the number of initial ground truth examples, and the number of labeled examples seem arbitrary. Given that cost-effectiveness is a primary motivation, it would have been more satisfying to see what the performance of, e.g., LLaMAA vs. finetuning if finetuning was performed on a number of samples equivalent in cost to the additional samples from the LLM.
>
> We totally understand your concerns. Regarding the choices of the number of gold data (ground truth) and the number of silver data (LLM-annotated), we follow the principle to minimize the two numbers separately.
> **As for # gold, we mainly go with a coverage consideration** (i.e. we want to include every type of entity for NER, each with a small amount of instances to be sampled from to decrease the randomness in sampling the demonstration set), and we aim that teacher LLM can reach a considerable performance with in-context examples provided.
> **As for # silver, the primary goal of LLMaAA is to induce a student model outperforms teacher LLM with minimal pseudo-labeled data**, and we find that 500 is sufficient for most NER and RE tasks, as shown by our results.
>
> To compare LLMaAA with FT with a number of samples equivalent in cost to the additional samples from the LLM, we must first identify how much the cost is. According to ([Wang et al., 2021](https://aclanthology.org/2021.findings-emnlp.354/)), the cost of human annotation is roughly estimated as \\$2/1k tokens, while ChatGPT charges for \\$0.002/1k tokens. Even taking the additional cost of template into account, we believe that human labeler can be 10x to 100x more expensive than LLM. Thus, 500 silver data is equivalent to dozens of gold data at most – even less than 100, which we evaluate for FT in Table 1 (LLMaAA only use silver for training student models and does not use gold).
>
> > C2: I am not sure I understand the motivation/value behind the "medium-data" setting studied in this paper. The datasets contain sufficiently many (hundreds to thousands of) examples not to be few-shot, but at the same time they are not big enough / sufficiently high quality that the downstream classifier's performance is near fully supervised BERT model.
>
> We want to clarify that we target at a semi-supervised setting, i.e. rich unsupervised data (text) with frugal gold data (demonstration/validation set): since in many real world scenarios we can see that unsupervised data is often abundant while annotation is the main bottleneck. In this case, gold data serves as the guideline for silver annotations (which requires equivalent efforts as prompt designing in PROMPTING), and unsupervised data serves as the proxy for the distillation from LLM to small models. We believe that such a setting is practical for applying LLM, and will reduce inference costs and latency for sensitive applications.
>
> As stated before, we want to build a student model outperforms teacher LLM (whose performance is satisfied but not excellent for applications) for online deployment, as a substitution. Though the performance has a gap to fully supervised BERT model, we believe that exceeding PROMPTING is still valuable. We will modify the introduction section to convey this message more clearly.
>
> > C3: I didn't find the argument in Section 6.2 particularly convincing.
>
> We mostly agree with your opinions. The motivation we add section 6.2 is that we find that the phenomenon (s > p) is interesting in the hard label distillation setting (no soft logits given), and as far as we concern no previous works have discussed the reason behind before. So we take an much relaxed setting to account for an intuitive understanding. We will revise this part and further add a discussion of limitations of our analyses.
>
> ### Questions:
>
> > Qa: I couldn't quite follow the description of how label verbalization is performed. Were label verbalizers from the cited works used or did you use a method from one of the cited works to create your own verbalizers?
>
> The label verbalizer we used is shown in Table 7, which is adapted from ([Lu et al., 2022](https://aclanthology.org/2022.findings-emnlp.490/)) with some modifications.
>
> > Qb: Are the labels provided by the LLM ever used in the set of examples drawn from for in-context learning?
>
> No, the labels are not used in a bootstrapping manner. It is an interesting direction, but we concern that such an approach may require multi time of inferences on the same sample, which contradicts the label efficiency issue that we care about.
>
> > Qc: In Table 1, the amount of data used for the PROMPTING baseline is '-', however, don't the prompts include in-context learning examples? Is this just the amount of data used by the downstram model? If so, I think it would be helpful to include the amount of data used by the LLM as well.
>
> In Table 1, the PROMPTING exactly denotes the performance of the teacher LLM in LLMaAA, and it select kNN examples from the same demonstration set (100 gold data) for NER (RE does not use in-context examples, as stated). Thanks for pointing out, and we will clarify it in the experiments section.
>
> > Qd: Similarly how many in-context examples are used for FewGen?
>
> It is also aligned with PROMPTING & LLMaAA (NER), and the examples are uniformly drawn from the same demonstration set. We find that we forget to mention that we use k = 5 for all experiments. We will add this information in the implementation section.
>
> ### Missing References, Typos Grammar Style And Presentation Improvements:
>
> Thanks for your instructive comments and careful suggestions. We will add a reference to ([Gao et al. 2023](https://openreview.net/forum?id=h5OpjGd_lo6)) in line 62-64, and discuss the two contemporary work in the related work section. We will also refine the expression (especially clarify efficiency and reliability), and modify Figure 1 in the revised version.

---

### Meta-Review · Area_Chair_NHDT · 2023-09-18

**Recommendation:** 2

**Metareview:**

This paper proposes LLMaAA using LLMs to annotate data in the active learning loop by optimizing both annotation and training processes to train task-specific models reliably. Reviewers appreciate that the method is reasonably well motivated and explained, presented with superior performance and comprehensive analysis with ablation study. They also bring up several areas for improvements: (1) The novelty is limited because a large portion of the framework directly borrows from existing techniques; (2) the lack of results on the effect dataset sizes, (3) improving the clarity and adding thorough analysis and discussions on efficiency and privacy. Overall, we strongly encourage authors to incorporate the feedback to improve the paper in the next version.

---

### Decision · Program_Chairs · 2023-10-07

**Decision:**

Accept-Findings

**Comment:**

This paper proposes LLMaAA using LLMs to annotate data in the active learning loop by optimizing both annotation and training processes to train task-specific models reliably. Reviewers appreciate that the method is reasonably well motivated and explained, presented with superior performance and comprehensive analysis with ablation study. They also bring up several areas for improvements: (1) The novelty is limited because a large portion of the framework directly borrows from existing techniques; (2) the lack of results on the effect dataset sizes, (3) improving the clarity and adding thorough analysis and discussions on efficiency and privacy. Overall, we strongly encourage authors to incorporate the feedback to improve the paper in the next version.